# Comparison of Fundamental Movement Skills and Physical Fitness in Hispanic and Non-Hispanic White Elementary School Children

**DOI:** 10.3390/healthcare11233028

**Published:** 2023-11-23

**Authors:** Hector Morales, Moises Velasquez, Samira Negrete, Louise A. Kelly

**Affiliations:** Department of Exercise Science, College of Arts and Sciences, California Lutheran University, Thousand Oaks, CA 91360, USA; hectormorales@callutheran.edu (H.M.); moisesvelasquez@callutheran.edu (M.V.); samiranegrete@callutheran.edu (S.N.)

**Keywords:** FITNESSGRAM^®^, minority youth, BMI percentile

## Abstract

Objective: Investigation of fundamental movement skills (FMS) and physical fitness differences among Hispanic and Non-Hispanic White elementary school children. Materials and Methods: This study included 194 children aged between 4 and 11 in kindergarten, 2nd, and 5th grades. For evaluable participants, *t*-tests were used to test potential differences in fitness or fundamental movement skills. Results: There were significant group differences across the participants in terms of age, height, body mass index (BMI), BMI percentile, arm circumference, and waist circumference. In addition, there were significant differences between Hispanic and Non-Hispanic White groups in terms of catching, running, dodging, punting, and two-hand striking (*p*’s < 0.05). Lastly, there were significant differences between Hispanic and Non-Hispanic White in pull-ups, push-ups, and sit-ups (*p*’s < 0.05). Conclusions: Our study shows Hispanic children have lower fundamental movement skills and physical fitness compared to their Non-Hispanic White peers. As poor fundament movement skills and physical fitness at such a young age may lead to adverse implications for their physical fitness and health in adulthood. Future studies should focus on increasing fitness, and FMS in Hispanic elementary school children.

## 1. Introduction

In the United States, obesity, especially childhood obesity, has been at the center of public health efforts, with nearly one-third of U.S. children and teens being overweight or obese [1]. A major concern about childhood obesity is that the condition is likely to continue into adulthood, with serious risks of related chronic conditions [2]. Similarly, diabetes and mental health impairment among children and adolescents are on the rise in the U.S., posing significant personal and social burdens [3]. Furthermore, there are serious health consequences that affect children without access to physical education, such as noticeable increases in weight and obesity [4]. Many ethnic minorities and children in low-income communities’ experience weight gain, leading to obesity [5]. Reversing the trend of pediatric obesity has proven difficult. While the need to decrease energy intake and increase energy expenditure has been widely accepted, interventions aiming to decrease intake have proven difficult, particularly in minority populations. Therefore, interventions that aim to increase energy intake by increasing physical activity may be more successful.

Physical inactivity and sedentary behavior are more prevalent among Hispanic children from low incomes [6]. Furthermore, Hispanic children exhibit a much higher rate of obesity than their White counterparts [7]. Children with lower levels of fundamental movement skills (FMS) are more likely to become physically inactive and to become obese as they grow. Accordingly, the prevalence of FMS proficiency among U.S. children is low, and children from disadvantaged backgrounds often rank the lowest in FMS proficiency [8]. Fundamental movement skills (FMS) are related to physical activity and are an important part of a child’s development. Many children become teenagers without having the right growth and development. Without these fundamental skills, children are at risk of serious health issues and impacts on their academic performance [9]. More research is needed to address the role of FMS in physical activity among young Hispanic children compared to Non-Hispanic White elementary school children.

FMS provide the foundation for a child’s physical and cognitive development, enabling them to function effectively in their daily lives [10]. In schools, young children have the best opportunity to practice and apply fundamental movement skills. FMS, including locomotor skills such as running, leaping, and sliding, are frequently developed in early childhood, along with object-control skills such as dribbling, throwing, and passing. Furthermore, interventions should be prioritized because children develop substantial skills, habitual behaviors, and motivation for active living during the early years of school [11]. In the United States, it is recommended that children receive 60 min of physical activity each day. Schools can be a valuable resource for Hispanic children to engage in physical activity and improve FMS growth.

The aim of this study is to investigate fundamental movement skills and physical fitness differences among Hispanic and Non-Hispanic White elementary school children.

## 2. Materials and Methods

### 2.1. Design and Subjects

The present cross-sectional study recruited children from Moorpark city in Ventura County in California, USA. According to the 2020 US Census data, Moorpark has a population of 36,284, with a population density of 1038.4/km^2^. The ethnic make-up of Moorpark is 59.4% White, 32.3% Hispanic/Latino, and 8.3% “other races”. Moorpark has a median household income of $107,820, which is considered higher than the national median income of $67,251; however, this is offset by the high living cost of California, US. Participants were pair-matched for age, gender, time of measurement (for season, school days/school holidays), and socio-economic status (SES). The paired design was intended to reduce the influence of factors other than ethnicity on the two groups and increase study power. All six elementary schools agreed to participate in the study, but only three were randomly selected. The study included 196 4- to 11-year-old children from kindergarten, 2nd, and 5th grades. All children from kindergarten, 2nd and 5th grades completed the same motors skills and physical fitness tests. Parental consent was obtained from all parents, and children consented voluntarily. California Lutheran Universities’ Institutional Review Board approved the study.

### 2.2. Anthropometric Measurements

Using a beam medical scale and wall-mounted stadiometer (SECA, Hamburg, Germany), we measured weight and height to the nearest 0.1 kg and 0.1 cm, respectively. Body mass index (BMI) and BMI percentiles for age and gender were determined using EpiInfo Version 1.1–2.0 (CDC, Atlanta, GA, USA). Anthropometric tape measures were used to measure waist and hip circumferences (to the nearest 0.5 cm). All anthropometric data were collected according to the Canadian Standardized Test of Fitness (CSTF) Operations Manual (Ottawa: Government of Canada, Fitness, and Amateur Sport, 1986).

### 2.3. Assessment of Fundamental Movement Skills

There were six fundamental motor skills evaluated: run, vertical jump, catch, overhand throw, forehand strike, and kick. These fundamental motor skills were evaluated using an 11-item fundamental movement skill test, which has been detailed elsewhere [12]. The assessment of fundamental movement skills was qualitative rather than quantitative because qualitative assessment methods are not influenced by student strength and can experience difficulty executing a fundamental motor skill. Research assistants rated the presence or absence of each of the qualitative components of each of these skills on four out of five trials [13]. A total of twelve research assistants were used in the study. The impact of involving more than one observer has been rigorously evaluated. The trained observer and the research assistant consistently achieved <4% inter-observer differences. Researchers who participated in the study of inter-observer agreement also participated in the current study.

### 2.4. Assessment of Physical Fitness

The FITNESSGRAM^®^ program 3rd addition was used to measure cardiorespiratory fitness and muscular strength. FITNESSGRAM^®^ is an assessment and promotion tool developed by the Cooper Institute (Dallas, TX, USA). It has been the recommended assessment tool and promotion tool by the Society of Health and Physical Educators (SHAPE) and is the required assessment for the state of California and many other states and districts. FITNESSGRAM^®^ has gained widespread support and adoption, largely due to its sound scientific basis [14,15]. The participants were asked to perform various fitness tests, including pushing themselves up 90 degrees, curling up for abdominal strength, sitting and reaching for flexibility, and running a mile for cardiovascular fitness assessment.

### 2.5. Assessment of Ethnicity

Using the self-reported questionnaires parents completed prior to inclusion in the intervention, the ethnicity of the children was determined. If the children were not of Hispanic origin, they were categorized into the Non-Hispanic White group. Children who were of Hispanic origin were only included in the study if both parents and grandparent were Hispanic. By doing so, a more accurate picture of the fundamental differences in movement skills between Hispanic and Non-Hispanic White elementary school children could be obtained.

### 2.6. Statistical Analysis

All data were checked for normality prior to statistical analysis. *T*-tests or Mann–Whitney U tests were used to assess differences in fitness and fundamental variables between Hispanic and Non-Hispanic White elementary school children. The characteristics of Hispanic and Non-Hispanic White were compared using the *p*-value to detect statistical differences in anthropometric, FMS, and fitness characteristics. Statistical analyses were performed using SPSS (Mac version 24.0) and an alpha of 0.05 for all statistics.

## 3. Results

### 3.1. Anthropometric Measurements

There were no significant group differences across the participants in weight and hip circumference. There were significant group differences across the participants in age, height, BMI, BMI percentile, arm circumference, and waist circumference (Table 1).

### 3.2. Fundamental Movement Skills Variables

Table 2 presents the movement skill outcomes for the evaluable participants. There were significant differences between Hispanic and Non-Hispanic White children across the groups in catching, run, dodge, punt, and two-hand strike (*p* < 0.05). There were no significant differences across the groups in kick, vertical jump, overhand throw, ball bounce, leap, and forehand strike (*p* < 0.05).

### 3.3. Physical Fitness Variables

Physical fitness outcomes for the evaluable participants are presented in Table 3. There were significant differences between Hispanic and Non-Hispanic White children across the groups in pull-up, push-up, and sit-up (all *p*’s < 0.05). There were no significant differences between Hispanic and Non-Hispanic White children across the groups in the trunk lift, sit/reach, and 1 mile run (all *p*’s > 0.05).

## 4. Discussion

The results of this study indicate that there are several key differences to emphasize among Hispanic and Non-Hispanic White elementary school children. Firstly, we observed ethnic differences in obesity, FMS, with Hispanic elementary kids being larger, and possessing weaker FMS. Secondly, we observed differences in fitness between both ethnicities, Hispanic children were less fit when compared with Non-Hispanic White children. FMS have been associated with predicting subsequent obesity from childhood to adulthood [16]. Therefore, it is important to develop these skills at a young age, as FMS provide underlying support for successful participation in physical activities and sports throughout lifetimes. Accordingly, preschool children with higher FMS competence are likely to have higher levels of physical fitness and activity as adults [17]. In addition, movement professionals must also have a broad understanding of the FMS development of Hispanic children [18]. With the expanding Hispanic population, as well as the health risks outlined, there is a high rate of sedentary behaviors and health risks associated with the Hispanic population [19]. Consequently, children from low-income and ethnic minority backgrounds are at greater risk of developmental delays. Accordingly, in Hispanic communities, the opportunities for promoting physical activity and reducing sedentary activities differ due to their different family and neighborhood backgrounds [20].

The findings of the study suggest that object-control skills such as catching, two-hand strike, and punting made a big difference as to whether or not the children were able to perform physical activity tasks such as pull-ups, push-ups, and sit-ups. According to previous research, higher levels of proficiency in object-control skills are associated with higher levels of physical activity [21]. This shows that the results of the object-control skills among the Hispanic and Non-Hispanic White children played important roles if they were able to perform well on fitness tests. Similarly, a more recent study found that object control skills predicted physical activity in both ethnic groups more strongly than all other sociodemographic factors put together [22]. Thus, the development of object control skills for physical activity promotion would also benefit young children.

Moreover, an interesting finding of the study was that there was a significant difference among Hispanic and Non-Hispanic White in the FMS of running, but there was not a significant ethnic difference in the 1 mile run fitness test. In similar research, previous results suggested that locomotor skills competency may be significantly associated with Non-Hispanic children’s physical activity, but not with Hispanic children’s activity [22]. Subsequently, this corresponds to the results portrayed here by displaying conflicting reports with locomotor skills among the children. Moreover, the findings of the study coincide with other research that proposes FMS and physical activity are related to each other in early school years regardless of sociodemographic background [23]. This shows that FMS and physical activity are significantly correlated among the children and may provide insight into what the children lack when it comes to FMS. Furthermore, several recent studies indicate that, once developed, FMS is likely to carry on throughout childhood and adolescence [22].

An important strength of this study is the relatively large number of randomly selected children who are representative of the general Moorpark population for this age group. Additionally, this study emphasized only using objective measures [24]; conversely, the existing literature has shown that only a few large studies have included objective measures of movement skills and physical fitness. Furthermore, several previous studies have reported benefits from interventions based on subjective outcomes, but this is probably due to a bias in the reporting of events by subjects and parents. One potential limitation of the study was the absence of data on what the participants among Hispanics and Non-Hispanic White did outside of school hours. This may have accounted for the increases in motor skill and fitness level in the Non-Hispanic White group. In addition, overweight status was measured using BMI percentile; a more accurate measure is needed to obtain a better depiction of anthropometric measurements. There is a paucity of normative data for fitness and FMS data for children from kindergarten to 4th grade. Standardized tests in California begin in 5th grade. Finally, to our knowledge, there is no ethnic specific fitness and FMS data for children in California, thereby making comparison of our data to normative impossible at the current time.

## 5. Conclusions

In conclusion, our study shows Hispanic children have lower fundamental movement skills and physical fitness compared to their Non-Hispanic White peers. The paucity of data on fitness and FMS levels in Hispanic children is of grave concern, as poor fundament movement skills and physical fitness at such a young age may lead to adverse implications for physical fitness and health in adulthood. Consequently, future studies should target interventions specifically to increase increasing fitness and FMS in Hispanic children.

Future studies should focus on increasing fitness and FMS in Hispanic elementary school children.

## Figures and Tables

**Table 1 healthcare-11-03028-t001:** Anthropometric Characteristics of Participants.

	Total Group (N = 194)		Non-Hispanic White (N = 97)		Hispanic (N = 97)		
	Mean	SD	Mean	SD	Mean	SD	*p*-Value ^1^
Age (years)	7.86	2.08	8.28	1.9	7.43	2.12	0.004 *
Height (M)	1.30	0.14	1.32	0.14	1.27	0.15	0.017 *
Weight (kg)	31.72	13.23	31.40	10.56	32.05	15.54	0.74
BMI Percentile)	64.19	29.20	58.92	27.53	69.45	30.03	0.015 *
Arm Circumference (cm)	20.03	3.83	19.34	2.98	21.06	4.68	0.005 *
Waist Circumference (cm)	60.43	11.11	58.36	8.41	62.53	13.00	0.009 *
Hip Circumference (cm)	71.00	11.15	70.70	8.80	71.31	13.12	0.71

^1^ *p* value—obtained on performing Mann–Whitney tests for difference between Non-Hispanic White and Hispanic children, * indicates significant difference between Non-Hispanic White and Hispanic children.

**Table 2 healthcare-11-03028-t002:** FMS Characteristics of Participants.

	Total Group (N = 194)		Non-Hispanic White (N = 97)		Hispanic (N = 97)		
	Mean	SD	Mean	SD	Mean	SD	*p*-Value ^1^
Catch	4.42	1.55	4.64	1.57	4.20	1.50	0.047 *
Kick	4.82	1.91	4.98	1.93	4.65	1.89	0.23
Run	3.23	1.33	3.40	1.35	3.06	1.29	0.071 *
Vertical Jump	2.99	1.34	2.96	1.41	3.02	1.26	0.75
Overhand Throw	3.45	1.79	3.55	1.89	3.36	1.68	0.47
Ball Bounce	2.51	1.43	2.56	1.62	2.45	1.21	0.62
Leap	2.56	1.37	2.56	1.52	2.56	1.21	1.00
Dodge	2.75	1.33	3.12	1.36	2.37	1.18	0.00 *
Punt	4.54	2.29	4.93	2.42	4.14	2.09	0.016 *
Forehand Strike	3.69	2.00	3.90	2.12	3.48	1.86	0.15
Two-Hand Strike	5.16	2.40	5.51	2.55	4.81	2.21	0.044 *

^1^ *p* value—obtained on performing Mann–Whitney tests for difference between Non-Hispanic White and Hispanic children, * indicates significant difference between Non-Hispanic White and Hispanic children.

**Table 3 healthcare-11-03028-t003:** Fitness Characteristics of Participants.

	Total Group (N = 194)		Non-Hispanic White (N = 97)		Hispanic (N = 97)		
	Mean	SD	Mean	SD	Mean	SD	*p*-Value ^1^
Trunk lift	30.02	8.21	30.50	9.52	29.63	6.97	0.52
Pull-up	1.41	2.63	2.02	3.26	0.91	1.87	0.011 *
Push-up	9.93	8.25	13.86	8.30	6.40	6.45	0.00 *
Sit-up	17.36	29.99	22.66	37.99	13.02	20.58	0.049 *
Sit/Reach	27.26	8.86	28.24	9.78	26.39	7.91	0.18
^1^ Mile Run	0.30	0.19	0.32	0.18	0.28	0.20	0.28

^1^ *p* value—obtained on performing Mann–Whitney tests for difference between Non-Hispanic White and Hispanic children, * indicates significant difference between Non-Hispanic White and Hispanic children.

## Data Availability

Data available upon written request to the corresponding author.

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
