# Peer review of "Comparison of Fundamental Movement Skills and Physical Fitness in Hispanic and Non-Hispanic White Elementary School Children"

_healthcare, 2023, doi:10.3390/healthcare11233028_

Round 1
Reviewer 1 Report
Comments and Suggestions for Authors
Thank you for your contributions to the field. I appreciated the opportunity to read this manuscript…I have identified some relatively minor corrections needed in the manuscript related to consistency, spelling, and grammar. Therefore, I provided some suggestions for corrections (please see attached PDF document).

Comments on the Quality of English LanguageAuthor Response
Response to reviewer 1.
Thank you for your very helpful comments. To make the re-review process easier for you we have highlighted the changes in red.
Overall
- Be consistent with the words that are capitalized or lower case when using the phrase “non-Hispanic white” as sometimes the N and/or the W are capitalized and other times they are not. Search the entire manuscript, including the captions and columns headings of the tables, for this phrase so it can be updated for consistency in all parts.
Our Response: thank you we have now make the corrections you suggested.
Title & Abstract
- Line 5: The number “1” at the end of the author line should be superscript.
Our Response: thank you we have now make the corrections you suggested.
Line 12: Should the word “activity” be changed to “fitness”?
Our Response: thank you we have now make the corrections you suggested.
Line 17: Delete “...using the p-value to detect statistical differences...”.
Our Response: deleted as suggested.
Lines 18-19: Delete the following sentence: “To identify any noticeable differences, the means and standard deviations of each group were compared.”
Our Response: deleted as suggested.
Lines 23 & 24: Change “(all p’s <0.05)” to “(p < 0.05)”
Our Response: thank you we have now make the corrections you suggested.
- Introduction
- Line 34: Change “obesity have” to “obesity.
Our Response: thank you, we have now made the correction you suggested.
Lines 41-43: Revise this sentence. Here is a possible revision: “Many ethnic minorities and children in low-income communities experience weight gain that leads to obesity.”
Our Response: thank you, we have now made the correction you suggested.
Line 34: Change “Children” to “children”
Our Response: thank you, we have now made the correction you suggested.
Line 51: change “...many children are growing into teenagers without...” to “many children become teenagers without...
Our Response: thank you, we have now made the correction you suggested.
Lines 64-65: Revise this sentence. Here is a possible revision: “Schools can be a valuable resource for Hispanic children to engage in physical activity and improve FMS.”
Our Response: thank you, we have now made the correction you suggested.
Line 67: Should the word “activity” be changed to “fitness”?
Our Response: thank you, we have now made the correction you suggested.
- Materials and Methods
- Lines 80-81: The percentages sum to over 100%. Is this because people indicated more than one ethnic group on the questionnaire? Double check these numbers to be sure they are correct.
Our Response: thank you, we have now made the correction you suggested.
- Line 83: Change the first comma to a semicolon.
Our Response: thank you, we have now made the correction you suggested.
- Line 90: Include the brand name (and manufacturer location) of the medical beam scale.
Our Response: thank you, we have now made the correction you suggested.
- Line 95: I think you need a parenthesis in front of Ottawa (e.g., “(Ottawa: Government of ....)”).
Our Response: thank you, we have now made the correction you suggested.
- Line 110: It says “..., described here”. Should there be a reference here?
Our Response: thank you we were referring to our study and have tried to clarify this in our manuscript.
- Line 116: The AAPHERD organization has changed its name and acronym to “Society of Health and Physical Educators (SHAPE)”.
Our Response: thank you, we have now made the correction you suggested.
- Line 121: Add the word “assessment” to the end of the sentence so it reads “...cardiovascular fitness assessment.”
Our Response: thank you, we have now made the correction you suggested
- Results
- Line 142: Should “Hip” be capitalized?
Our Response: thank you, no it should not, we have now made the correction you suggested.
Line 143: Include Table 1 at the end of the sentence so it reads “...circumference (Table 1).”
Our Response: thank you, we have now made the correction you suggested.
Lines 144-145: Delete this sentence: “Table 1 presents the anthropometric outcomes for the evaluable participants.” 4. Line 171 & 173: Change “(all p’s <0.05)” to “(p < 0.05)”.
Our Response: thank you, have deleted the sentence and corrected the p values as requested.
Table 2: center the column heading and the values in the cells of the “p-value” column. Also, in Table 3 (Mean, SD). 6. Table 2 and Table 3: The column heading “p-value” should be “p-value1” to match with Table 1.
Our Response: thank you, corrected tables as requested.
- Tables 1-3: The first row of Table 3 separates “Total group”, Non-Hispanic White, and Hispanic categories but Tables
Our Response: thank you, corrected tables as requested.
Reviewer 2 Report
Comments and Suggestions for Authors
First of all, I would like to thank the editors of the journal for the possibility of reviewing this article that talks about the differences in FMS between Hispanic and non-Hispanic children. Although it is something that is commonly studied, I believe that it is important to develop studies that affirm or refute this theory over the years, given that it is something that, as the authors mention, if the social and economic conditions of the country change, the population could be affected.
Regardless, I would like to clarify a few issues regarding the manuscript.
ABSTRACT. Typo errors like the bold in methods should be solved previuous to publish.
INTRODUCTION.
The phrase between lines 39 and 41 needs to be explained better since I don't understand the meaning.
The introduction does not follow a clear thread, I recommend developing the ideas further, including more previous articles that talk about the same thing and relating everything to each other.
The second objective mentioned is not developed throughout the text, nor does it present results. Therefore I think it should be eliminated. "In addition, a novel 8-week teacher-led physical activity intervention is being examined to determine the physiological health effects and improvement of motor skills in Hispanic children".
METHODS.
The Methods section shows a very detailed description of the Moorpark census. If you are going to focus so much on that municipality, I suggest adding all this data in the introduction.
Have the scales and measurement elements used been validated? If so, provide references.
RESULTS. The results seem well presented and very clear to me but I would like the economic variable to have also been measured since the introduction mentions it as possibly responsible for the differences.
DISCUSSION. The discussion is well argued but it seems to me that a more similar article should be compared.
CONCLUSION. The conclusion must be written based on the results of your study, therefore it cannot include references. Another section would be that of future studies, but at least a conclusion on the own results must be written.
Finally, I encourage the authors to make these changes because they seem easy to solve, since the study is well designed and the results are well collected.
Author Response
Thank you for your very helpful comments. To make the re-review process easier for you we have highlighted the changes in red.
ABSTRACT. Typo errors like the bold in methods should be solved previous to publish.
Our Response: thank you, corrected as requested.
INTRODUCTION.
The phrase between lines 39 and 41 needs to be explained better since I don't understand the meaning.
Our Response: apologies, we have amended the sentence to be more clear.
The introduction does not follow a clear thread, I recommend developing the ideas further, including more previous articles that talk about the same thing and relating everything to each other.
Our Response: thank you, we really appreciate your feedback, however, we also are having difficulty finding previously published articles (other than those we cited) that look at these ethnic differences. While there are manuscripts that look at fitness and FMS they do not investigate these in Hispanic children. This ethnic group is the fastest growing ethnic minority group in the US and in particular California and yet, there is a paucity of research.
The second objective mentioned is not developed throughout the text, nor does it present results. Therefore, I think it should be eliminated. "In addition, a novel 8-week teacher-led physical activity intervention is being examined to determine the physiological health effects and improvement of motor skills in Hispanic children".
Our Response: thank you, removed the second objective as requested.
METHODS.
The Methods section shows a very detailed description of the Moorpark census. If you are going to focus so much on that municipality, I suggest adding all this data in the introduction.
Our response: Thank you for your comment, we are not focusing on Moorpark, we just wanted to explain the sample we recruited and show that they were representative of the community.
Have the scales and measurement elements used been validated? If so, provide references.
Our response: We have added a reference for the FITNESSGRAM® program, but we are unaware of any study that has validated the 11 item fundamental movement skill test.
RESULTS. The results seem well presented and very clear to me but I would like the economic variable to have also been measured since the introduction mentions it as possibly responsible for the differences.
Our Response: Unfortunately, we do not have an economic variable as the school district we were recruiting from felt collecting this information was “too personal” for the parents. In lieu of economic variables we thought it best to describe the median income of the sample.
DISCUSSION. The discussion is well argued but it seems to me that a more similar article should be compared.
Our response: thank you for your positive comment; we agree that similar articles would be beneficial, however, there are no other articles that we know of that are like ours. In the US, Research in the Hispanic community, especially in the pediatric population is severely lacking, especially as they are the fasted growing ethnic minority group in the US. They are also the highest for risk of obesity and metabolic disorders. We are hoping that our manuscript will add to the paucity of data that is published on this topic.
CONCLUSION. The conclusion must be written based on the results of your study, therefore it cannot include references. Another section would be that of future studies, but at least a conclusion on the own results must be written.
Our response: thank you, we have amended our conclusion as requested.
Reviewer 3 Report
Comments and Suggestions for Authors
Comments in the attachment

Author Response
- The purpose of the work is: to examine basic motor and physical skills activity differences between Latino and non-Hispanic white elementary school children. Additionally, a novel, 8-week teacher-led physical activity intervention is examined to determine physiological health effects and improvements in motor skills in Latino children - results related to the second part of the study's objectives are not included in the paper.
Our Response: another reviewer asked us to remove the second objective from our paper, so we have subsequently done so.
2.The material and methods state that children aged 4-11 were examined. It is not known whether the research methods were the same between age groups.
Our Response: thank you, we have added the following line into our study design section.
All children from kindergarten, 2nd and 5th grades completed the same motors kills and physical fitness tests.
3.BMI - the authors write in the introduction and in the discussion about the problem of obesity, and in the tables they showed BMI = total - 18.25. Non Hispanic White 17.6 and Hispanic 18.91. According to publicly available tables, these are results indicating underweight or normal weight.
Our Response: thank you, these BMI’s are correct, but only for adults. We do not use raw BMI to classify overweight/obesity status in the pediatric population as this can be misleading. If we use the Hispanic children’s BMI as an example 18.91, it will classify them as being normal weight. But when we plot 18.91 against age and gender, we see they have a percentile of >64, which means they are heavier than 64% of the children at their age and gender. BMI percentiles are used in children in the US to determine a more accurate picture of weight status, like using SDS scores in the United Kingdom.
- When analyzing inter-individual differences, it is necessary to separate groups characterized by the same age and sex, otherwise the analyzes are questionable.
Our response: we chose to see if there was a difference in how our research staff were scoring the FMS as accurately as possible as we did not want any difference to be attributed to testing error.
- In the discussion, the obtained results should be compared to norms and standards to determine the extent to which the respondents differ from the general public. Intergroup differences alone are insufficient. Therefore, I would recommend correlating "national" norms with your own results in your work results.
Our response: we have added the following to our paper in an attempt to address your comment:
There is a paucity of normative data for fitness and FMS data for children from kindergarten to 4th grade. Standardized tests in California begin in 5th grade. Finally, to our knowledge, there is no ethnic specific fitness and FMS data for children in California, thereby making comparison of our data to normative impossible at the current time.
- In the conclusion, you should write about the results of your work, and not refer to the literature on the subject. Postulates for other researchers should be outside the conclusion because they do not prove the current work.
Our response: thank you, we have now amended our conclusion based on your suggestion.
- The proposed physical fitness tests are an advantage, but age, gender and "national" standards should also be taken into account here.
Our response: we agree, that is why we pair matched each of the participants. With regards to national standards, CA does not have standards for this age group. Standards only begin in 5th grade.
8.In general, to determine whether a given environment differs from each other, the obtained results should be verified with other standardized results, because it is known from the work that both groups examined differ from each other, but it is not known whether this is good or bad.
Our response: We can see what the review is referring too. However, as previously mentioned it is impossible for us to verify if our sample is similar in there results to standards as no standards are available in CA for this age group. What we do know is that white non-Hispanic children have higher physical activity levels compared to Hispanic children. But we have a paucity of data on fitness and FMS.
Round 2
Reviewer 3 Report
Comments and Suggestions for Authors
I am satisfied with the corrections and the authors' response. I propose to remove the results related to BMI from Table 1, leaving BMI (Percentile) or to cleary explain why this sreults was included in the work.
Author Response
Thank you for reviewing our paper again. We have removed the BMI as requested from table 1.